# Effect of Silica Nanoparticles in Xanthan Gum Solutions: Evolution of Viscosity over Time

**DOI:** 10.3390/nano12111906

**Published:** 2022-06-02

**Authors:** Dayan L. Buitrago-Rincon, Véronique Sadtler, Ronald A. Mercado, Thibault Roques-Carmes, Lazhar Benyahia, Alain Durand, Khalid Ferji, Philippe Marchal, Julio A. Pedraza-Avella, Cécile Lemaitre

**Affiliations:** 1Grupo de Investigación en Fenómenos Interfaciales, Reología y Simulación de Transporte (FIRST), Universidad Industrial de Santander, Bucaramanga 680002, Colombia; dayan.buitrago-rincon@univ-lorraine.fr (D.L.B.-R.); ramerca@uis.edu.co (R.A.M.); 2Laboratoire Réactions et Génie des Procédés, Université de Lorraine, CNRS, F-54000 Nancy, France; veronique.sadtler@univ-lorraine.fr (V.S.); thibault.roques-carmes@univ-lorraine.fr (T.R.-C.); philippe.marchal@univ-lorraine.fr (P.M.); 3Institut des Molécules et Matériaux du Mans, Le Mans Université, CNRS, F-72085 Le Mans, France; lazhar.benyahia@univ-lemans.fr; 4Laboratoire de Chimie-Physique Macromoléculaire, Université de Lorraine, CNRS, F-54000 Nancy, France; alain.durand@univ-lorraine.fr (A.D.); khalid.ferji@univ-lorraine.fr (K.F.)

**Keywords:** xanthan gum, silica nanoparticles, colloidal dispersions, rheology, aging tests

## Abstract

The effect of silica nanoparticles (NP-SiO_2_) in xanthan gum (XG) solutions was investigated through the analysis of viscosity profiles. First, hydrocolloid XG solutions and hydrophilic NP-SiO_2_ suspensions were characterized individually through rheological measurements, with and without salt (NaCl). Then, nanofluids composed of XG and NP-SiO_2_ dispersed in water and brine were studied through two different aging tests. The addition of nanoparticles was shown to produce a slight effect on the viscosity of the fresh fluids (initial time), while a more remarkable effect was observed over time. In particular, it appears that the presence of NP-SiO_2_ stabilizes the polymer solution by maintaining its viscosity level in time, due to a delay in the movement of the molecule. Finally, characterization techniques such as confocal microscopy, capillary rheometry, and Zeta potential were implemented to analyze the XG/NP-SiO_2_ interaction. Intrinsic viscosity and relative viscosity were calculated to understand the molecular interactions. The presence of NP-SiO_2_ increases the hydrodynamic radius of the polymer, indicating attractive forces between these two components. Furthermore, dispersion of the nanoparticles in the polymeric solutions leads to aggregates of an average size smaller than 300 nm with a good colloidal stability due to the electrostatic attraction between XG and NP-SIO_2_. This study proves the existence of interactions between XG and NP-SiO_2_ in solution.

## 1. Introduction

Due to their practical significance, natural polymers, such as xanthan gum (XG) continue to attract the attention of researchers. Xanthan gum has been widely used for many applications in the food, pharmaceutical and petroleum industries among others. In particular, XG is of great interest for Enhanced Oil Recovery (EOR) processes used in the petroleum industry.

Water-soluble polymers, such as XG, are used in EOR to increase the viscosity of the water that is injected into the reservoirs in order to extract the oil. Indeed, this viscosity increase enhances the mobility ratio between the crude oil and the polymer solution, which impeaches viscous fingering and leads to a greater oil recovery [1,2,3,4]. Polymers such as xanthan gum induce high viscosities in aqueous solutions, even at low concentration. However, EOR not only requires formulations of high viscosity, but also that this property remains stable in time against temperature conditions, ionic forces, and shear [5,6].

Xanthan gum is an anionic heteropolysaccharide produced by the microorganism *Xanthomonas campestris*. The primary structure of xanthan gum is a linear, linked β-D-glucose backbone with a trisaccharide side chain on every other glucose at C-3. The polymer chain consists of a D-glucuronic acid unit between two D mannose units. One-half of the terminal D-mannose unit contains a pyruvic acid residue linked via keto group to the 4 and 6 positions, with an unknown distribution. The D-mannose linked to the main chain contains an acetyl group at position O–6 (Figure 1) [7].

In aqueous solutions, the addition of polymer causes an increase of the viscosity, due to inter- and intramolecular forces within the polymer and with the surrounding solvent molecules. Furthermore, the presence of charged functional groups in its side chain (carboxyl groups (-COOH)) makes it an anionic polysaccharide [8]. These polymer solutions exhibit a shear-thinning rheological behavior (the viscosity decreases with increasing shear rate).

Xanthan macromolecules may exhibit two different conformational structures: an ordered and a disordered (helical) conformation [9,10,11]. The XG undergoes conformational changes in aqueous solutions when exposed to temperature, salinity, and deformation stress [11]. As a result, the molecule rotates its conformation and the intermolecular and intramolecular bonds existing in the system, tend to weaken and then break, causing a viscosity loss and/or polymer precipitation [2].

In recent years, silica nanoparticles (NP-SiO_2_) have been used as an innovative technique to address different industrial challenges [12,13,14]. Nanoparticles in general, and especially NP-SiO_2_, display exceptional properties owing to their small size and large surface area. The surface of NP-SiO_2_ allows an interaction with the (-OH/H) functional groups present in the xanthan structure, leading the polymer chains to link with the NP-SiO_2_. Recent studies have suggested a limitation of the xanthan conformational transition due to the addition of NP-SiO_2_ [15]. In these nanofluids, the interaction of the polymer with the NP-SiO_2_ allows new cross-linked structures, which delay the conformational transition, improving the properties of XG solutions [11,16]. However, xanthan gum-silica nanoparticle interactions in dilute solutions have been little studied.

In a previous work, the polymeric dilution curve of xanthan gum was elaborated by sectorizing the regions of diluted zone and semi-diluted zone and different methods to prepare the nanofluid (XG and NP-SiO_2_ in water or brine) were evaluated [17]. This work is extended here, to describe the evolution of the viscosity of nanofluids and the effect of NP-SiO_2_ over time. The objective of the present study is to investigate and understand the effect of NP-SiO_2_ on xanthan gum solutions and to study the interactions in this colloidal system. Rheological and aging measurements are employed to characterize the viscosity and the stability of the fluids in time. Intrinsic and relative viscosity, Zeta potential, and confocal microscopy measurements are used to characterize the synergistic interactions between the polymer and the nanoparticles.

## 2. Materials and Methods

### 2.1. Materials

The xanthan gum was purchased from Sukin Industries. The silica nanoparticles were fumed Aerosil 300 amorphous hydrophilic (300 ± 30 m^2^/g) from Evonik Industries. Sodium chloride was supplied by Sigma-Aldrich 99.5% (58.44 g/mol). Deionized water was used as solvent and glutaraldehyde, provided by Sigma-Aldrich (50 wt.% in H_2_O), was added as a bactericidal agent.

### 2.2. Preparation of Xanthan Gum Solutions and Nanoparticles Suspensions

The polymer solutions were prepared with two different solvents. The first solvent was deionized water, in which polymer powder was incorporated at different concentrations (150, 300, 600, 1200, 2400 and 4800 ppm). The second solvent was brine, previously prepared with different concentrations of NaCl (0.5, 1 and 3wt%), in which the polymer powder was incorporated at a concentration of 600 ppm, to evaluate the influence of ionic forces on the viscous behavior of the fluid. All solutions were left under magnetic stirring at room temperature, and bactericide was added to the solvent at a fixed concentration (80 ppm) before incorporating the polymer. In order to ensure that the polymer was fully hydrated, two different hydration times were tested: 24 and 48 h. The viscosity profiles after these two durations were compared.

Similarly, nanoparticle suspensions were prepared with the same two solvents, before their rheology was characterized. On the one hand, nanoparticles were dispersed at different concentrations (100, 200, and 300 ppm) in deionized water. On the other hand, NaCl brines were prepared with different concentrations (0.5, 1, and 3wt%), and then, the nanoparticles were added at a concentration of 300 ppm, so that the variability of the viscosity of the nanosuspensions under different ionic charges was evaluated. All the suspensions were left under magnetic stirring for 1 h at room temperature.

### 2.3. Influence of Temperature during Nanofluids Preparation

In previous work, various preparation methods of these nanofluids were evaluated [17]. The protocols differed in the polymer hydration time, the order of addition of the components, the sonication time applied to the NP-SiO_2_ suspensions, and the interaction time between the components. The method yielding the highest viscosity profile (30.1% higher than the XG solution without NP-SiO_2_) with a minimum number of steps was retained. It consists in incorporating the XG and the NP-SiO_2_ at the same time in the solvent and to leave the resulting mixture under magnetic stirring during 24 h. In the same study, the concentration regimes for XG solutions were determined: the dilute regime corresponds to 0–600 ppm XG, and the semi-dilute regime to 600–4800 ppm XG [17].

In the present work, the effect of temperature applied during nanofluid preparation was evaluated. Using the previously selected preparation method, the operating temperature was set successively to 20 °C, 40 °C, 60 °C, and 80 °C. The formulations were prepared with fixed components concentrations (XG: 600 ppm; NP-SiO_2_: 300 ppm and 3wt% NaCl) and at constant pressure.

### 2.4. Rheological Characterization

Two different apparatuses were used to measure the viscosity of the different fluids considered in this study: a rheometer and a capillary viscometer.

A strain-imposed Rheometric Scientific RFS II rheometer was used, equipped with a rheoreactor geometry. This geometry consists of a rotating helical ribbon of 17.95 mm radius rotating in a tank of 27.5 mm internal radius and 42 mm inner height. It is adapted to low viscosity fluids since it generates higher torques than classical geometry, like plate/plate cells. Furthermore, the ribbon agitator ensures the homogeneity of the mixture. The accuracy of this system has been proven by previous studies [18]. The tests were executed at constant temperature of 25 °C or 60 °C. The shear rate was varied between 0.1 and 100 s^−1^, with sensitivities of 1 Pa·s for the viscosity and 0.002 N·m for the torque. Finally, the effect of NP-SiO_2_ on the stability of the viscous profile of the formulations under aging conditions was evaluated. All measurements were repeated at least twice for each formulation to confirm reproducibility. The particular shear rate of 7.3 s^−1^ was used to compare some results since it is a representative rate for the flow profile under well conditions for EOR applications [19].

Capillary viscometers measure a viscosity from the time it takes for a volume of polymer solution to flow through a thin capillary.
(1)$$t_{s}\sim \frac{\mu_{s}}{\rho_{s}}$$
(2)$$t\sim \frac{\mu }{\rho }$$
with $t_{s}$ and t the flow times obtained for the solvent and the considered solution, respectively, $\rho_{s}$ and ρ the solvent and solution densities, and $\mu_{s}$ and μ the solvent and solution viscosities. Considering equivalent densities (since the solutions are dilute), the relative viscosity can be obtained through
(3)μrel=tts

In order to evaluate the evolution hydrodynamic radius of the polymer in time with or without nanoparticles, the intrinsic viscosity [μ] of the formulations was determined from rheological measurements. The intrinsic viscosity of fresh fluids (initial time) was obtained from capillary viscosimetry, while the evolution of the intrinsic viscosity over time was calculated from the rheograms obtained with the rheometer at a shear rate of 7.3 s^−1^.

Intrinsic viscosity is related to the hydrodynamic volume occupied by isolated individual polymer molecules [20]. The relationship between dilute polymer solution viscosity and concentration can be described by many empirical forms, the most common being the Huggins equation [21].
(4)$$\frac{\mu_{sp}}{C}=\left[\mu \right]+k^{\prime }{\left[\mu \right]}^{2}C$$
where $k^{\prime }$ is the Huggins constant and C is the polymer concentration. The specific viscosity $\mu_{sp}$ is defined as the relative increase of viscosity.
(5)$$\mu_{sp}=\frac{\mu -\mu_{s}}{\mu_{s}}=\mu_{rel}-1$$
with μ and $\mu_{s}$, the apparent viscosities of the solution and the solvent, respectively, and $\mu_{rel}=\mu /\mu_{s}$ the relative viscosity. The Huggins equation may be simplified by removing the second-order term for very dilute solutions. Combining the simplified Equations (4) and (5), the relative viscosity reads [22].
(6)$$\mu_{rel}=1+\left[\mu \right]C$$

In this study, [μ] was determined for all fluids of different ages by measuring the slope of the relative viscosity as a function of polymer concentration, within the range 1.0 < *μ_rel_* < 2.0. The obtained intrinsic viscosities were then plotted as a function of aging time [23].

### 2.5. Aging Tests

In the context of EOR applications, it is important to consider the viscosity stability of XG solutions over time. Indeed, these fluids may remain several months in the reservoir rock and should keep their efficiency.

In order to study the evolution of the formulation viscosity over time, two aging tests were performed on the nanofluids. The first test was conducted for an aging time of 1 month. Fluids of different concentrations of NaCl (0% and 3wt%), NP-SiO_2_ (0 ppm and 300 ppm) and a fixed concentration of 1000 ppm XG (semi-diluted zone) were stored at a constant aging temperature (60 °C) for this duration.

A second longer test was performed for an aging time of 7 months. The tested fluids contained NaCl (3wt%), polymer at a dilute (400 ppm XG) or semi-diluted (1000 ppm XG) concentrations and nanoparticles of different concentrations (0 ppm and 300 ppm NP-SiO_2_). The viscous profile of the fluids was measured as a function of the shear rate at different times.

### 2.6. Confocal Microscopy

Confocal microscopy measurements were performed using a ZEISS LSM800 Confocal Laser Scanning Microscope. The microscope was equipped with four solid-state lasers, three GaAsP PMT detectors, and one T-PMT detector for transmission light detection. Images were obtained for nanoparticle suspensions (300 ppm), polymer solutions (1000 ppm), and nanofluids (XG: 1000 ppm; NP-SiO_2_: 300 ppm) in fresh conditions (initial time) and aged conditions (1 month). Confocal microscopy enables the creation of sharp images without any disturbing fluorescent light.

### 2.7. Zeta-Potential Measurements

Zeta potential measurements of polymer solutions in water and brine (3wt% NaCl), and nanofluid (400 ppm XG, 300 ppm NP-SiO_2_ and 3wt% NaCl) were carried out with a Malvern nanosizer at 25 °C. The Zeta potential reflects the potential difference between the electric double layer of electrophoretically mobile particles and the layer of dispersant around them at the slipping plane. It provides an estimation of the surface charge of the particles.

## 3. Results and Discussion

### 3.1. Viscosity Behavior of Xanthan Gum Solutions

The viscosity curves of the polymer solutions were compared for hydration times of 24 h and 48 h. The viscosities obtained after 24 h of stirring were very close to that obtained after 48 h. It was thus concluded that a 24 h stirring period is enough to ensure the complete hydration of the polymer. The polymer solutions show a marked shear-thinning behavior for concentrations above 600 ppm, Figure 2. For lower concentrations (150 ppm and 300 ppm XG), shear-thinning is weak, and rheological behavior is close to Newtonian. As expected, the viscosity increases with polymer concentration.

Xanthan gum solutions in brines of different NaCl concentrations were then considered. For polyelectrolytes solutions, it was evidenced that an increase of ionic strength causes a significant decrease in viscosity for all the solutions, with a permanent shear-thinning behavior, see Figure 3. At a representative shear rate of 7.3 s^−1^, the viscosity decreases by 61% with a concentration of 3% NaCl compared to a saltless water polymer solution. Such behavior is due to the conformational transition of xanthan. In water, the xanthan molecule is extended due to electrostatic repulsion of the negatively charged acetyl groups [24]. When salt is added, charge screening causes the side chains to collapse down to the backbone, hence giving the xanthan molecule a rod-like shape, more compact, corresponding to a reduced hydrodynamic size of the molecule, leading to a reduced viscosity [25,26].

### 3.2. Viscosity Behavior of Nanoparticle Suspensions

The viscosity behavior of NP-SiO_2_ suspensions was analyzed to explore the degree of dispersion of the nanoparticles. Indeed, for a smaller agglomerate size or a better dispersion of the NP-SiO_2_, the suspensions tend to be more viscous and to adopt a Newtonian behavior (viscosity independent of the shear rate). However, the flow profiles obtained in the present study for three different NP-SiO_2_ concentrations did not show a significant difference, see Table 1. Indeed, there was no significant viscosity change between suspensions of 100 ppm to 300 ppm NP-SiO_2_. This is possibly due to the low studied concentrations.

The effect of salt on the NP-SiO_2_ suspensions rheology was then investigated. No significant difference was observed in water or in brine. The viscosity profiles were not altered at all. However, since for EOR applications, working brines are estimated to contain an average concentration of 30,000 ppm (3%) NaCl, this salt concentration is chosen as a reference in the present study.

### 3.3. Influence of Temperature during Nanofluids Preparation

After choosing the preparation method from a previous study [17], the effect of temperature imposed during the preparation of nanofluids was evaluated at 20, 40, 60 and 80 °C. The intermediate concentration of 600 ppm XG, separating the dilute and semi-dilute regimes, was selected and the NP-SiO_2_ concentration was fixed to 300 ppm. All measurements were then performed at the same temperature 60 °C.

The obtained viscosities are shown in Table 2 for a shear rate of 7.3 s^−1^, and compared to the viscosity of the polymer solution without nanoparticles. A viscosity increase ranging between 15.7 and 21% is obtained for the lower tested temperatures (20 to 60 °C), showing an effect of the presence of the particles. The highest viscosity is obtained for a preparation temperature of 20 °C. At higher temperatures (80 °C), however, the measured viscosity is almost that of the polymer solution, which indicates that the effect of the NP-SiO_2_ is negligible.

This low effect of temperature could be related to the ionic strength. Indeed, the rheological behavior results from a balance of several factors. Hydrating the polymer at high preparation temperatures promotes thermal disassociation of the polymer structure. In aqueous solutions, the backbone of xanthan changes from a disordered (or partly ordered in the form of a randomly broken helix) to an ordered structure when increasing the temperature. The conformation of the polymer in the solution mainly affects the viscosity of the fluid in that a more disordered conformation will achieve a higher viscosity value. In this case, by increasing the preparation temperature, the molecule is ordered and the interaction with the nanoparticles is promoted. However, the high ionic strength (3% NaCl) could inhibit the polymer chain extension/repulsion at high temperature due to charge screening effects and stabilize the ordered conformation, which leads to a lower fluid viscosity [10].

### 3.4. Aging Tests—Stability

In order to evaluate the stability of the fluid viscosity in time, aging tests were carried out. Short aging tests (1 month) were applied to polymer solutions (1000 ppm XG) and nanofluids (1000 ppm XG and 300 ppm NP-SiO_2_), elaborated in water or brine (3wt% NaCl). The temperature used both for storage and for the viscosity measurements was 60 °C (Figure 4).

The formulations containing NP-SiO_2_ present higher viscosities than polymer solutions without nanoparticles, at all times. Furthermore, the viscosity of all formulations is found to decrease in time, but the viscosity loss is limited in the presence of nanoparticles. In distilled water at 20 °C, the backbone of xanthan is disordered (helical form), and highly extended due to the electrostatic repulsions from the charged groups on the sidechains [10]. This disordered conformation of the polymer could promote a greater synergistic XG/NP-SiO_2_ interaction. Formulations prepared in a disordered conformation of the polymer (without NaCl) display a higher viscosity than salty solutions, due to the fact that the xanthan chains are more repelled from each other. A greater repulsion increases the hydrodynamic radius of the molecule, which then occupies a larger volume within the fluid leading to a higher viscosity of the system. These results show the influence of nanoparticles on the conformation of XG, and consequently, on the viscosity of polymer solution, showing a greater XG/NP-SiO_2_ interaction.

The effect of the nanoparticles on the viscosity of the fluid is more remarkable in solutions with water solvent. At a shear rate of 7.3 s^−1^ for fresh fluids (time zero), the viscosity is increased by 35% with nanoparticles in water, compared to the 20% increase obtained in brine. The viscosity stability over time is also different in water and in brine. Solutions prepared in distilled water undergo a greater loss. After 30 days, a viscosity loss of 45% is measured for polymeric solutions and only 30% for nanofluids. In brine (3wt% NaCl), the effect of nanoparticles on fluid viscosity is reduced: the viscosity hardly changed during the entire aging period. In the presence of NaCl, the polymer stabilizes to an ordered conformation. Xanthan gum chains are much more rigid and stable due to the collapse of the trisaccharide side chains onto the main backbone. This effect reduces the mobility of the molecule and hinders the degradation of the molecular chains with or without nanoparticles [27].

A second aging test was performed for a longer time (7 months) to further study the effect of the nanoparticles and their interaction with the polymer in solutions with high ionic strength (3wt% NaCl, Figure 5). For these tests, two polymer concentrations were considered (dilute, 400 ppm XG and semi-dilute, 1000 ppm XG) and a single nanoparticle concentration (300 ppm NP-SiO_2_). The storage temperature of the fluids and the test measurement temperature were the same, 60 °C.

After 7 months, all the polymer solutions showed a strong decrease in viscosity while the nanofluids displayed a moderate viscosity loss. The same behavior was observed for both dilute and semi-dilute polymer solutions. The semi-dilute polymer solution without nanoparticles underwent a 72% viscosity reduction, while the same formulation containing nanoparticles exhibited a viscosity drop of only 34%. For the dilute polymeric solution, a viscosity drop of 73.5% in time was found without nanoparticles, while the drop was halved (37%) in the presence of nanoparticles. These longer tests confirm that the presence of NP-SiO_2_ promotes the stability of the fluid, maintaining the viscous profile.

These rheological results could be attributed to weak physical attractive forces between the components, such as hydrogen bonds. The surface of silica nanoparticles exhibits silanol groups (Si-OH) while XG chains contain hydroxyl groups (-OH), which makes (−O−H^δ+^…O^δ−^) dipole–dipole interactions possible [28]. However, the nature of the interaction between xanthan gum and silica nanoparticles is still an open question. This aspect will be discussed in the following sections.

### 3.5. Intrinsic and Relative Viscosity

Intrinsic viscosity of xanthan gum solutions and nanofluids is shown in Figure 6 as a function of aging time. Using the simplified Huggins equation, Equations (4)–(6) were applied for each month and compared the solutions without NP-SiO_2_ (polymer solutions) and with NP-SiO_2_ (nanofluids).

In dilute solutions, the polymer chains are separate, and the intrinsic viscosity of a polymer in solution depends only on the hydrodynamic dimensions of the polymer (hydrodynamic volume) [29]. The intrinsic viscosity of XG solutions with and without NP-SiO_2_ is shown in Figure 6 as a function of aging time. The estimated intrinsic viscosity (41.52 dL/g) obtained for the nanofluid is much higher than that obtained from polymer solution (37.41 dL/g). These results suggest that the hydrodynamic radius of xanthan gum increases because of the nanoparticles, and this increase of the hydrodynamic volume suggests an efficient interaction between the polymer and the nanoparticle. For both polymer solutions and nanofluids the intrinsic viscosity decreases over time. However, the reduction of the polymer dimensions is more remarkable in the absence of nanoparticles.

In order to evaluate the interaction between the xanthan gum and the silica nanoparticles, the relative viscosity $\mu_{rel}$ is plotted as a function of xanthan concentration (Figure 7). As expected, the relative viscosity $\mu_{rel}$ of polymer solutions and nanofluid was found to increase with polymer concentration. The slope is higher for nanofluids than for polymer solutions, indicating a higher intrinsic viscosity and thus a higher polymer hydrodynamic radius in the presence of nanoparticles. Furthermore, a linear variation was found for the polymer solutions, which denotes a mass contribution of the xanthan gum without specific interactions [10]. On the contrary, the relative viscosities of the nanofluids did not vary linearly, indicating that specific attractive forces were present between the xanthan molecules and the silica nanoparticles. The non-linear trend is due to another factor than the polymer concentration increase. Thus, the hypothesis of intermolecular binding between xanthan gum and silica nanoparticles is supported by the intrinsic viscosities and relative viscosities values obtained for the different formulations.

### 3.6. Confocal Microscopy

The colloidal dispersion of the nanoparticles in the brine and the polymeric solution was observed through confocal microscopy tests, see Figure 8. All images were unchanged for all formulations after 1 month aging. The sensitivity of the confocal microscope is around 300 nm, below this scale it is not possible to observe. The images of the nanoparticle suspensions, Figure 8a, showed microscopic agglomerates reaching sizes up to 20 μm. As expected, the polymer molecules were not visible on the images of polymer solutions, see Figure 8b. On nanofluids images, nanoparticle aggregates were not distinguishable, see Figure 8c. This means that the aggregation of the nanoparticles is affected by the polymer, decreasing the size of the NP-SiO_2_ aggregates from 20 μm to sizes less than 300 nm, which is the maximum resolution of the equipment.

This indicates that an interaction between xanthan gum and silica nanoparticles exists, which leads to the breaking of micron-sized nanoparticles agglomerates. 

### 3.7. Zeta Potential

Aqueous and saline solutions of xanthan gum, as well as nanofluids containing xanthan gum and NP-SiO_2_ in brine, were also characterized by Zeta potential measurements (Table 3). Xanthan gum is a polyelectrolyte that exhibits a negative surface charge in aqueous solution due to its anionic character provided by the carboxyl groups present in the trisaccharide side chain of its repeat unit [8].

Measurements show that the addition of nanoparticles in the polymer solution induces an increase of the zeta potential. The Zeta potential of the nanofluids shows stronger interaction between the XG and NP-SiO_2_, indicating that improved dispersion stabilization has been achieved [30]. This high negative value for the nanofluids contributes to explain the stability of the Xantham/NP-SiO_2_ suspensions over time.

Polymeric solutions of Xanthan gum in distilled water obtained ζ of −19.1 mV, confirming the electrostatic repulsion between the polymer chains. By adding the NaCl ions to the polymer solution, ζ increases to −14.4 mV due to the screening of the Na^+^ and Cl^−^ ions on the XG-charged monomer units, resulting in a decrease in the surface charge density [31]. On the other hand, when NP-SiO_2_ is added to xanthan gum solution in brine, the surface electric charge drops down to −23.0 mV. This decrease is probably related to the formation of hydrogen bonds between NP-SiO_2_ and XG, which are responsible for the formation of stronger networks and structures. As a result of these physical interactions, the XG repeat units are less exposed to the undesired protective effects of salt ions.

## 4. Conclusions

In the present work, the interaction between xanthan gum (XG) macromolecules and silica nanoparticles (NP-SiO_2_) in water or brine was investigated. Various polymer, nanoparticle and salt concentrations were considered. Different experimental techniques were used: rheometry, capillary viscosimetry, confocal microscopy and Zeta potential measurements. Aging tests were performed, during which the fluids were stored for several months at 60 °C. Measurements were carried out just after fluid preparation and at successive aging times.

The individual components were first characterized in water and brine. XG molecules are satisfyingly dissolved after 24 h and the resulting solutions (without particles) exhibits a shear-thinning behavior. The addition of salt to these solutions causes a viscosity decrease while maintaining a shear-thinning behavior. Nanoparticle suspensions (without polymer) are Newtonian for the considered particle concentrations and their viscosity is not affected by the addition of salt.

The preparation temperature of the nanofluids (XG and NP-SiO_2_ in brine) was varied, and a moderate effect was found on the resulting viscosities.

The rheology of the XG solutions with and without NP-SiO_2_ was characterized over time. The presence of nanoparticles led to higher viscosities. The viscosity of all fluids decreased in time, but the decrease was limited by the presence of nanoparticles, which demonstrates the stabilization of xanthan gum solutions by the addition of nanoparticles. However, these positive effects of NP-SiO_2_ are attenuated by the presence of salt.

The intrinsic viscosity of the polymer solutions with and without nanoparticles was calculated from the rheological measurements. The intrinsic viscosity is higher with NP-SiO_2_ at every time. This indicates that the hydrodynamics radius of the polymer is larger when nanoparticles are present, which is a sign of polymer/particle interaction. Furthermore, the relative viscosity was found to vary nonlinearly with polymer concentration in the presence of NP-SiO_2_, which is another sign of polymer/particle interaction.

Confocal microscopy images show that the nanoparticles dispersed in water form large aggregates (20 microns), but that the aggregates are much smaller (under 300 nm) in the presence of xanthan gum, evidencing an action of polymer on particle dispersion. It was also shown experimentally that the presence of nanoparticles affects the Zeta potential of XG solutions.

The nature of the interactions between the XG macromolecules and the silica nanoparticles remains an open question. It is suggested that they may result from weak interactions by hydrogen bonds.

## Figures and Tables

**Figure 1 nanomaterials-12-01906-f001:**
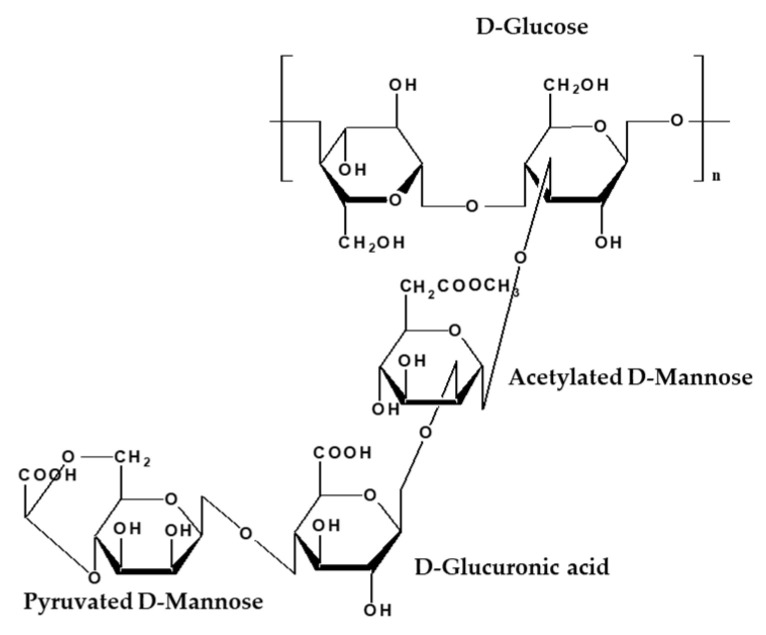
Chemical structure of xanthan gum.

**Figure 2 nanomaterials-12-01906-f002:**
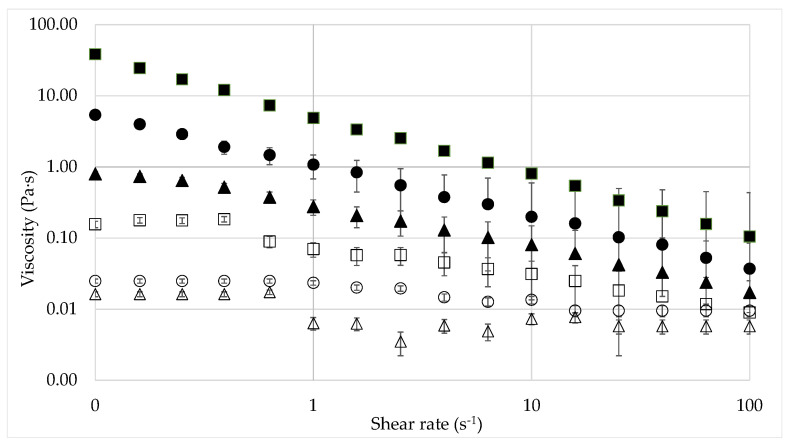
Viscosity of aqueous xanthan gum solutions of different concentrations as a function of shear rate. Hydration time 24 h, 4800 ppm XG (■), 2400 ppm XG (●), 1200 ppm XG (▲), 600 ppm XG (□)., 300 ppm XG (○), and 150 ppm XG (Δ).

**Figure 3 nanomaterials-12-01906-f003:**
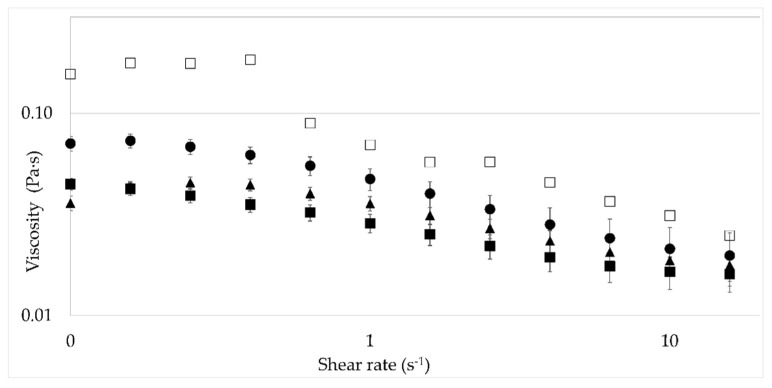
Viscosity of polymer solutions (600 ppm XG) with different NaCl concentrations as a function of the shear rate. Water (□), 0.5% NaCl (●), 1% NaCl (▲), and 3% NaCl (■).

**Figure 4 nanomaterials-12-01906-f004:**
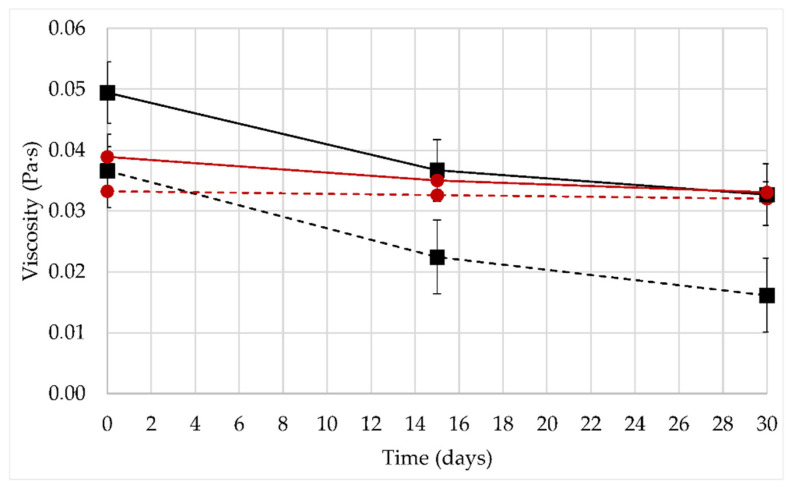
Viscosity as a function of time. Viscosities were measured at a shear rate of 7.3 s^−1^. Storage and measurements performed at T=60 °C. Nanofluid in water (1000 ppm XG and 300 ppm NP-SiO_2_) (–■). Polymer solution in water (1000 ppm XG) (- -■). Nanofluid in 3% NaCl brine (1000 ppm XG and 300 ppm NP-SiO_2_) (–●). Polymer solution in 3% NaCl brine (1000 ppm XG) (- - ●).

**Figure 5 nanomaterials-12-01906-f005:**
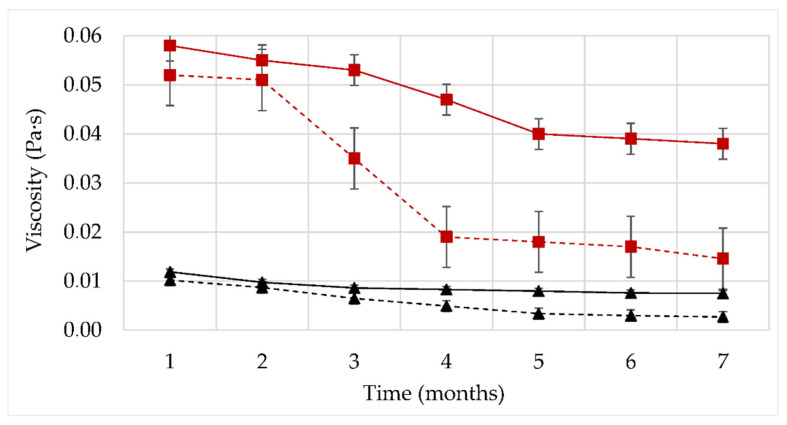
Viscosity as a function of time at a shear rate of 7.3 s^−1^. Storage and measurement at T=60 °C. Semi-dilute nanofluid (1000 ppm XG, 300 ppm NP-SiO_2_ and 3% NaCl) (–■). Semi-dilute polymer solution (1000 ppm XG and 3% NaCl) (- -■). Dilute nanofluid (400 ppm XG, 300 ppm NP-SiO_2_ and 3% NaCl) (–▲). Semi-dilute polymer solution (400 ppm XG and 3% NaCl) (- -▲).

**Figure 6 nanomaterials-12-01906-f006:**
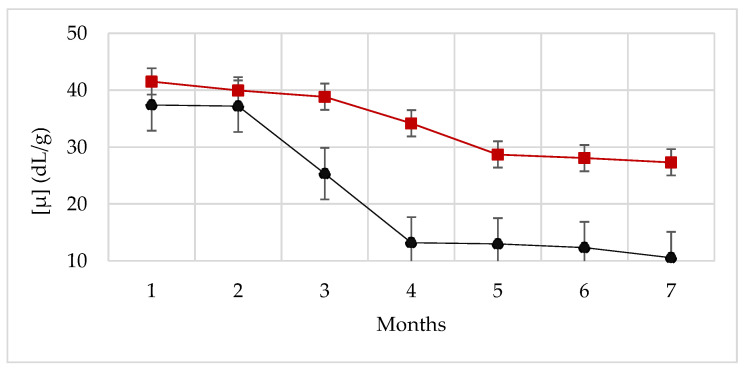
Intrinsic viscosity [μ] as a function of time. Storage and tests performed at T=60 °C. Nanofluid (–■). Polymer solution (–●).

**Figure 7 nanomaterials-12-01906-f007:**
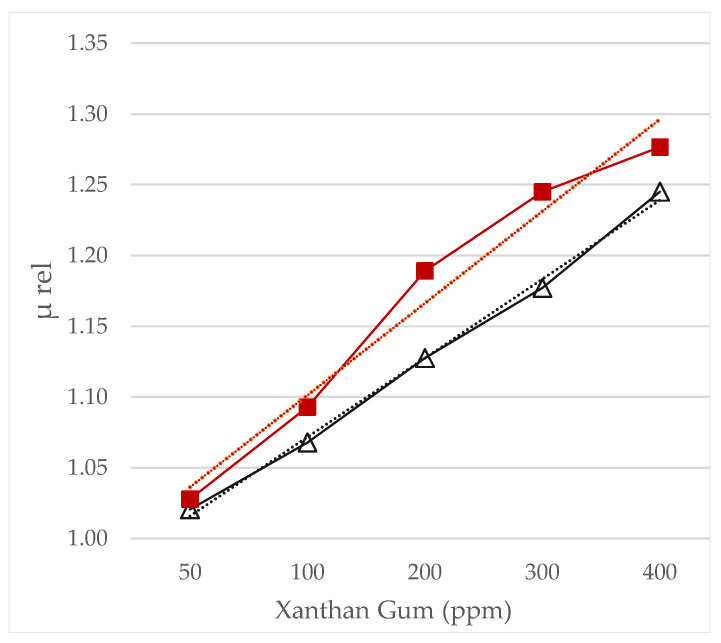
Relative viscosity *μ_rel_* against xanthan gum concentration. Nanofluid (–■). Polymer solution (–Δ).

**Figure 8 nanomaterials-12-01906-f008:**
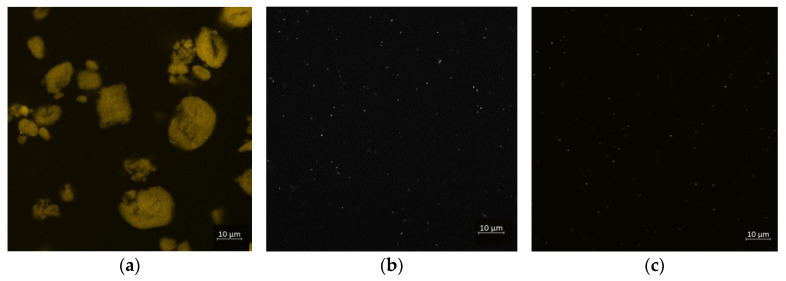
Confocal microscopy images. (**a**) Nanoparticle suspension: 300 ppm NP-SiO_2_ and 3% NaCl in deionized water; (**b**) polymer solution: 1000 ppm XG and 3% NaCl in deionized water; (**c**) nanofluid: 1000 ppm XG, 300 ppm NP-SiO_2_ and 3% NaCl in deionized water.

**Table 1 nanomaterials-12-01906-t001:** Viscosity of nanoparticles suspensions.

Fluid	Viscosity (Pa·s) ^1^
Water	0.001000
100 ppm NP-SiO_2_	0.001012
200 ppm NP-SiO_2_	0.001140
300 ppm NP-SiO_2_	0.001167

^1^ Viscosity—Newtonian behavior.

**Table 2 nanomaterials-12-01906-t002:** Relative viscosity increases due to preparation temperature.

Fluid	Preparation Temperature	Vicosity Pa·s	Viscosity Increase ^1^
Polymer solution	20 °C	0.030604	Control sample
Nanofluid	20 °C	0.037030	21.0%
Nanofluid	40 °C	0.035488	15.7%
Nanofluid	60 °C	0.035959	17.5%
Nanofluid	80 °C	0.031124	1.7%

^1^ Viscosity at a shear rate 7.3 s^−1^.

**Table 3 nanomaterials-12-01906-t003:** Zeta potential of XG solution in water or brine and nanofluid (XG/NP-SiO_2_/NaCl).

Formulation	Solvent	Zeta Potential (mV) ^1^
Xanthan gum (400 ppm)	Water	−19.1
Xanthan gum (400 ppm)	Brine 3% NaCl	−14.8
Nanofluid: XG (400 ppm) and 300 ppm NP-SiO_2_	Brine 3% NaCl	−23.0

^1^ Temperature 25 °C.

## Data Availability

The data presented in this study are available on request from the corresponding author.

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
