# Peer review of "Effect of Silica Nanoparticles in Xanthan Gum Solutions: Evolution of Viscosity over Time"

_nanomaterials, 2022, doi:10.3390/nano12111906_

Round 1

Reviewer 1 Report

The manuscript is well written. Taking into account the wide use of xanthan gum and silica for the stabilization of several systems, the interest for this research may be broad. The main problem I find in this study is that the rheological characterization is not entirely complete. Flow properties (which can be better discussed) do not provide all the information that may be necessary or important in these types of systems. A characterization of viscoelastic properties is therefore missing. However, taking into account the number of results shown and their analysis, I believe that improving the manuscript in its current version could be susceptible to be published taking into account its interest for readers. 

Comments and questions:

  • Names of microorganisms should be italicized
  • Several units must be corrected (Pa.s, pa.s, s-1 ...)
  • If possible, adding color to figures facilitates their analysis
  • The fitting parameters for adjusting the flow curves and their analysis should be included
  • A statistical analysis is necessary in some figures, for example Figure 3. Apparently, and more taking into account the scale used, there is no significant difference between samples.
  • Why hasn't the viscoelastic properties of the systems been analyzed?
  • I do not doubt that the reusltados shown are the ones that were really seen, but in figures 8b and 8c nothing is seen or contributed.

Reviewer 2 Report

Write SiO2 with 2 in subscript all along the manuscript
write s-1 with « -1 » in superscript in every table or figure legends, and in the whole text.
Write  β-D-glucose and not β-d-glucose
The description of xanthan is too simplistic in the introduction to explain the phenomena that the authors highlight in this manuscript. Please describe more precisely Xanthan structure (with a general stucture for example) so authors will be able to explain, or try to explain, what happened during the formulation. Describe the possible interactions of SiO2 and Xanthan and aqueous solvent.

Reviewer 3 Report

The paper investigated the interaction of silicate nanoparticle (NP-SiO2)  and polymer xanthan gum (XG). The viscosity of XG solution with and without NP-SiO2 were tested and its change in time was studied. Characterization techniques such as confocal microscopy, capillary rheology, and Zeta potential were implemented to analyze the XG/NP-SiO2 interaction. The confocal microscopy result (Fig. 8) is very interesting and it showed that the interaction between XG and NP can significantly affect the size of NP. The manuscript can be better if the follow issues were considered.

  • I do not understand why “There was no particle precipitation of nanoparticles in the polymer solutions and no sedimentation takes place in all range of NP-SiO2 concentrations (100 ppm - 300 ppm)”. Can you explain it more clearly?
  • Please check the Fig.5 caption. The semi-dilute and dilute were not in consistent with “(dilute (400 ppm XG) and semi-dilute (1000 ppm XG))”.
  • Figure 3 is not clear.
  • I did not understand “Little change in the viscosity of the fluids is observed, an increase in the preparation temperature does not increase the viscosity of the fluid”. Do it imply that when preparation temperature is increased for other polymer or nanofluids, the viscosity will increase?
  • Viscosity value may be given in Table 2.
  • Are the data in Figure 1 and Figure 2 from experiments? It too linear to be believable.
  • Why viscosity unit of Pa.s was used? It is so big.
  • Is Figure 8 taken in the scale?
  • What is the size of nanoparticles? 20 micrometers? Does nanoparticle size affect the interaction between XG and NPs?
  • What are the interaction between XG and NPs? New cross-linked structures formed ? or ?

Round 2

Reviewer 1 Report

Taking into account the results obtained, I consider that the manuscript can be published in Nanomaterials in its current version